# An Analysis on the Effectiveness of Nitrogen Oxide Reduction from Applying Titanium Dioxide on Urban Roads Using a Statistical Method

Sang-Hyuk Lee, Jong-Won Lee, Moon-Kyung Kim and Hee-Mun Park *

Korea Institute of Civil Engineering and Building Technology, Gyeonggi-do 10223, Korea; slee@kict.re.kr (S.-H.L.); asca28@kict.re.kr (J.-W.L.); moonkyung@kict.re.kr (M.-K.K.)

* Correspondence: hpark@kict.re.kr; Tel.: +82-91-910-0323

**Abstract:** The purpose of this study was to analyze the effect of titanium dioxide ($TiO_2$) on reducing nitrogen oxide (NOx) concentrations using the statistical method of the Anderson-Darling test. To compare and analyze this effect, a spray-type form of $TiO_2$ was applied to the asphalt pavement surface on urban roads. Data acquisition for NOx concentration was collected from a test section with $TiO_2$ applied and a reference section without $TiO_2$ applied. The probabilities of occurrence of the NOx concentration in the test and reference section were estimated and compared using the Anderson-Darling test. In sum, most of the NOx concentrations were probabilistically lower in the test section. The average probability of the NOx concentration in the test section in the 'low' range was 46.2% higher than in the reference section. In the 'high' and 'moderate' ranges, the average probability of the NOx concentration compared to that of the reference section was lower by 28.1% and 18.8%, respectively. These results revealed that the photochemical reaction from the $TiO_2$ material applied on asphalt pavement was effective in reducing NOx.

**Keywords:** titanium dioxide; nitrogen oxide concentration; Anderson-Darling test; photocatalyst

## 1. Introduction

The steady growth in economic conditions and the rapid increase in the economic growth of neighboring countries in the past several decades have led to an accelerated degree of air pollution in South Korea. Most notably among air pollution issues is the widely discussed social issue regarding the increase of particulate matter concentrations to a serious level due to migration from neighboring countries and seasonal effects.

Particulate matter is defined by the US EPA as a complex of extremely small particles and liquid droplets. Particulate matter can be generally classified into two categories: particulate matter with less than 10 μm in diameter ($PM_{10}$) and particulate matter with less than 2.5 μm in diameter ($PM_{2.5}$), which are called fine particulate matter. Sources of particulate matter are known to include the combustion process at manufacturing industries, especially industrial plants that contribute more than 60%, while roads and non-point pollution sources, energy industry combustion and other production processes are known to each take up about 20%, respectively. Also, air pollutants emitted into the atmosphere can be the source of secondary pollutants that are further converted into particulate matter through chemical reactions [1]. $PM_{2.5}$ consists of pollutants containing heavy metals, sulfur dioxide ($SO_2$), nitrogen oxide (NOx), lead (Pb), ozone ($O_3$), and carbon monoxide (CO). They can be generally produced through chemical reactions that combine $SO_2$ and NOx from fossil fuel combustion and vehicle emissions with water vapor and ozone. NOx compounds can negatively affect human health, and the nitric acid and nitrate formed by the oxidation of NOx are the main sources of nitric acid rain that can have a negative effect on ecological environments [2]. Since NOx is a main source of generating $PM_{2.5}$ on urban roads, the technology development for reducing NOx concentration is crucial.

Recently, a method using titanium dioxide ($TiO_2$) has been proposed to reduce NOx concentration from air pollutions, and a number of studies have been carried out to remove NOx by applying $TiO_2$ on urban road structures and highway areas [3]. However, few studies analyzing the NOx reduction effect from applying $TiO_2$ on operating roads or highways have been carried out. Also, previously conducted studies had limitations in analyzing the reduction amount by measuring the NOx concentration at a certain point of time after applying $TiO_2$ to the roads. Therefore, an analytical method is needed to better understand the NOx reduction effect of $TiO_2$ material.

The objective of this study is to evaluate NOx reduction efficiency using field measurement data and statistical analysis after applying $TiO_2$ on the pavement surface of urban roads. A liquid type of $TiO_2$ material was applied on the asphalt pavement surface at the test section. The reference section is defined as a section without $TiO_2$ applied for relative comparison analysis of NOx reduction effect. NOx concentration data was monitored in the test and reference sections simultaneously for ten days. In order to analyze the NOx reduction effect of $TiO_2$ on urban roads, probability distributions for NOx concentration were estimated using the statistical method of the Anderson-Darling test. Anderson-Darling test is a statistical goodness-of-fit verification method presented as a criterion for reducing errors in the process of estimation. This Anderson-Darling test is a useful method for accurate prediction and estimation of the NOx reduction effect of $TiO_2$ through stochastic analysis. Therefore this method proposed in this study can carry out more accurate quantitative and qualitative analysis compared to existing methods. In addition, the occurrence probability of NOx concentration was calculated using the probability density function for the estimated probability distribution to analyze the NOx reduction effect in the test section.

## 2. Related Works

Since the late 20th century, many studies related to the occurrence of NOx, particulate matter, and other air pollutants on urban roads have been actively conducted. In the 1970s, lots of studies focused on the direct effects of traffic characteristics on the occurrence of NOx, $SO_2$, CO, and $O_3$ [4,5]. In the 1990s, several studies were performed on the short-term effects of harmful emissions. According to the research of Katsouyanni et al., an increase of 50 mg/m$^3$ in $SO_2$ level was associated with a 3% increase in mortality, and a 2% increase in $PM_{10}$ concentration [6].

In the early 2000s, some studies were carried out to investigate the long-term effects of harmful emissions. At the time, the World Health Organization (WHO) announced that NOx from main roads is a major source of air pollution (40–50%). In addition, there are 30% more NOx generated at road facilities such as tunnels, bridges, and highways compared to general roads [7]. Also, the European Environmental Agency published that exposure to NOx was responsible for around 70,000 deaths in the European Union and approximately 400,000 premature deaths were caused by exposure to high particulate matter concentration levels [8]. As such, particulate matter formed by NOx compounds from roads has a fatal effect on human health.

As part of a study on the perception and awareness of particulate matter pollution caused by urbanization and industrialization, Srimuruganandam and Nagendra sampled the 24-h average ambient particulate matter ($PM_{10}$ and $PM_{2.5}$) concentration at two intersections in Chennai, India, from November 2008 to April 2009 [9]. The study results show that the 24-h average ambient particulate matter concentration was significantly higher in the winter and monsoon season than in the summer. The results also indicated that secondary particulate matter from traffic conditions (22.9% in $PM_{10}$, 42.1% in $PM_{2.5}$) was the major cause of higher particulate matter concentrations in the study area.

Jandacka and Durcanska measured particulate matter fraction concentrations ($PM_{10}$, $PM_{2.5}$, $PM_{1.0}$), substance concentration, and crystals at 6 different monitoring stations in Zilina, Slovakia, from 2017 to 2019 to find sources of particulate matter in urban areas [10]. The study results revealed that characteristics of particulate matter is dependent on mea-

suring stations and seasons. In particular, higher concentrations of particulate matter were measured due to differences in the concentration of chemical elements contained in particulate matter on the roads. This variation of particulate matter concentrations was caused not only by traffic volumes but also by weather conditions.

Photocatalytic oxidation is a useful and effective method to remove NOx precursors. Therefore, many studies have been conducted recently with a focus on reduction of NOx concentrations using photocatalytic oxidation [11–14].

Folli et al. examined the reduction of NOx concentration after applying 100 mm thick double-cast concrete blocks containing $TiO_2$ in Copenhagen, Denmark [15]. This study presented that the average daily NO concentration in concrete blocks with $TiO_2$ was below 40 ppvb. Also, the NO reduction efficiency was higher in the summer than in the winter, and monthly average NO concentration in the photocatalytic section was about 22% lower than reference section during the summer. In particular, data measured around noon in the summer indicated that the average NO reduction efficiency is higher than 30% due to higher solar radiation.

Bocci et al. applied an asphalt emulsion and cement mortar with $TiO_2$ on the main and emergency lane of highway sections in Italy to analyze the applicability of photocatalyst material [16]. The effectiveness of photocatalytic treatments was evaluated by measuring NO reductions after 1, 17, 46, 88, 218, and 527 days from applying the proposed materials. The results show that all the products were effective after 24 h from the application with a 40% reduction efficiency for asphalt emulsion-based $TiO_2$ products and 23% reduction efficiency for cement mortar-based $TiO_2$ products. However, the reduction efficiency started to decrease gradually with elapsed time and the NO reduction efficiencies for asphalt and concretes products were 1.77% and 0.49% after 527 days, respectively. Moreover, the NO reduction efficiency of the main lane at a highway section was lower compared to the emergency lane at a highway section due to heavy vehicle traffic. The study also found that cement mortar-based $TiO_2$ products had faster performance degradation compared to asphalt emulsion-based $TiO_2$ products.

Wang et al. developed a new method for constructing photocatalytic air-purifying asphalt pavements and evaluated the feasibility and performance of new method [17]. This study adopted two ways to produce the $TiO_2$ modified aggregate, which were aggregate surface coating and pore filling. They investigated and compared the durability in terms of photocatalytic efficiency and mechanical performance under vehicle tire polishing. Both aggregate surface coating and pore filling methods produced satisfactory NO reduction rates around 40% before vehicle tire polishing but reduction efficiency was reduced in both methods after vehicle tire polishing. Comparing these two methods for NO reduction efficiency revealed that the pore filling method provided a better long-term NO reduction efficiency than aggregate surface coating method.

Yu et al. evaluated the NOx degradation performance of nano-$TiO_2$ as a coating material for the road environment using the photocatalytic test system designed by this study under various radiation intensities [18]. The material used in this study were anatase nano-$TiO_2$, activated carbon powder, silane coupling agent and deionized water. Using the presented materials, the impact of varying amounts of coating material and silane coupling agent were evaluated. The results show that the material has acceptable photocatalytic degradation performance and the proper amount of silane coupling agent could enhance the bonding performance of the material and asphalt mixture. For the roadside coating, sodium dodecylbenzene sulfonate was selected as the surfactant to carry out the photocatalytic degradation experiment of $NO_2$ with different dosages of surfactant. The results showed that when the mass ratio of nano-$TiO_2$ and surfactant was about 1:2, the catalytic degradation effect of the material was the best.

Ma et al. applied $TiO_2$ nanoparticles to asphalt pavements to conduct an experimental investigation to purify vehicle emissions of asphalt and asphalt mixtures modified by nano-$TiO_2$ [19]. The results revealed that the $TiO_2$ nanoparticle was beneficial to increase the viscosity and reduced the temperature sensitivity which would enhance its high-

temperature stabilization capability of asphalt. It was also found that the incorporation of nano-$TiO_2$ in mixtures significantly enhance the rheometeric properties of asphalt, the high temperature anti-rutting capacity and the anti-cracking properties, as well as water stabilization. Moreover, it verified that the nano-$TiO_2$-modified asphalt mixture possesses a positive impact on photocatalytic degradation of CH and NOx.

## 3. Understanding Generation and Reduction Mechanism of Pollutant

### 3.1. Converting Nitrogen Oxide to Particulate Matter

Fine particulate matter ($PM_{2.5}$) is mainly formed by secondary organic aerosols (SOA) and secondary inorganic aerosols (SIA), which are generated from physical and chemical reactions between sulfur oxide (SOx), nitrogen oxide (NOx), volatile organic compounds (VOCs), ammonia ($NH_3$), and ozone ($O_3$) [20,21]. However, a detailed formation process and transformation mechanism of SOA has yet to be established, especially one that takes into account external factors such as nitrogen emissions from natural soils (biogenic soil nitrogen emission), regional ozone generation, nitrogen monoxide and nitrous oxide emission from natural forests, and temperature and climate conditions [22,23].

SOAs are formed through three main processes [24]. First, semi-volatile gas-phase products generated from reactions between atmospheric oxides (hydroxyl radical (OH), nitrate radical ($NO_3$), and ozone) move into the particulate phase through gas-particle partitioning. Second, large oligomer particles with molecular weights ranging from 100 to 1000 form through acid catalyzation on the surface of existing aerosols. Third, water-soluble organic compounds (WSOCs) in the atmosphere dissolve into cloud/mist droplets or fine particulate matter containing moisture and then move into the particulate phase (aqueous chemistry including OH reactions, and acid catalyzation).

SIAs are generated by chemical reactions of gas-phase precursors such as NOx, $SO_2$, and $NH_3$ [21,24,25]. Ammonium nitrate ($NH_4NO_3$) and ammonium sulfate (($NH_4)_2SO_4$) are the typical SIAs in the atmosphere, and ammonium nitrate is mainly generated by NOx emissions from mobile pollution sources dominant on roads during the daytime [23,26]. Generation of $NH_4NO_3$ during the daytime begins with NOx oxidation [27]. Nitrogen monoxide, emitted from NOx emission sources such as mobile pollution, combines with ozone in the atmosphere to form nitrogen dioxide and reacts with hydroxyl radicals in the atmosphere to form nitric acid ($HNO_3$). Then, the nitric acid combines with ammonia to form ammonium nitrate.

### 3.2. Photocatalysis

The utilization of $TiO_2$ on pavement and road facilities has a significant effect on reducing NO concentration, a primary source of fine particulate matter. $TiO_2$ has a valence and conduction band, and the energy difference between the two bands (band gap energy). The band gap energy of $TiO_2$ in its anatase phase, which is mainly used as a photocatalyst, is 3.2 eV. When $TiO_2$ is applied on pavement and road facilities, the exposure to ultraviolet ray (UV) light greater than the band gap energy results in the formation of electrons ($e^-$) and holes ($h^+$). Those electrons and holes react with oxygen ($O_2$) and water ($H_2O$) in the air to generate active oxygen (superoxide anion ($O_2^{\cdot-}$)) and hydroxyl radical ($OH\cdot$) on the $TiO_2$ surface. Those products break down NOx into nitrates ($NO_3^-$), reducing NOx in the atmosphere when they wash out as aqueous nitric acid ($HNO_3\cdot$) solution in the rain. Moreover, nitrates ($NO_3$) can be removed through biological denitrification in groundwater [28]. The formulas for photocatalytic activity by ultraviolet rays described above are as follows [29]:

$$TiO_2 \xrightarrow{UV} h^+ + e^- \tag{1}$$

$$O_2 + e^- \rightarrow O_2^{\cdot-} \tag{2}$$

$$OH^- + h^+ \rightarrow OH \tag{3}$$

$$O_2^{\cdot-} + H^+ \rightarrow HO_2^{\cdot} \tag{4}$$

The process of removing the superoxide anion ($O_2^{\cdot-}$) and hydroxyl radical (OH·) generated from reactions with NOx are as follows:

$$NO + OH\cdot \ \rightarrow \ HNO_2 \tag{5}$$

$$HNO_2 + \ OH\cdot \ \rightarrow \ NO_2 + \ H_2O \tag{6}$$

$$NO + \ HO_2\cdot \ \rightarrow \ NO_2 + \ OH \tag{7}$$

$$NO_2 + \ OH\cdot \ \rightarrow \ HNO_3\cdot \tag{8}$$

$$NO_x + \ O_2^{\cdot-} \ \rightarrow \ NO_3^- \tag{9}$$

The hydroxyl radical (OH·) generated from the photocatalytic reaction in Equation (3) then reacts with NO and $NO_2$ according to Equations (5) and (6) to finally form nitric acid ($HNO_3\cdot$). Nitric acid ($HNO_3\cdot$) is water-soluble, so it can be easily removed from the photocatalytic surface by ambient air environments such as rain. Superoxide anion ($O_2^{\cdot-}$) generates water ($H_2O\cdot$), which in turn reacts with NO as detailed in Equation (7) to finally produce $HNO_3\cdot$ as shown in Equation (8). Also, NO molecules react with superoxide anion ($O_2^{\cdot-}$) to produce nitrates ($NO_3^-$), which is effective in removing NOx [30].

## 4. Methodology

### 4.1. Characteristics of TiO$_2$ Material

Photocatalysts are substances that use light as an energy source to conduct catalytic reaction and applied in various areas for antibacterial, deodorizing and degradation of harmful substances. There are various types of photocatalysts including titanium dioxide ($TiO_2$), zinc oxide (ZnO), cadmium sulfide (Cds) and tungsten trioxide ($WO_3$) etc. ZnO and Cds generate harmful Zn and Cd ions by causing the catalyst itself to decompose by light. $WO_3$ has the disadvantage of having partial efficiency for specific substances. Since $TiO_2$ is stable chemically and physically and is not eroded by acids, bases, or organic solvents, it is widely used as a construction material [28]. Therefore, $TiO_2$ with anatase crystals was selected as photocatalyst for NOx reduction for urban roads in this study. Figure 1 show a transmission electron microscopy image and the energy dispersive spectrum for $TiO_2$ material.

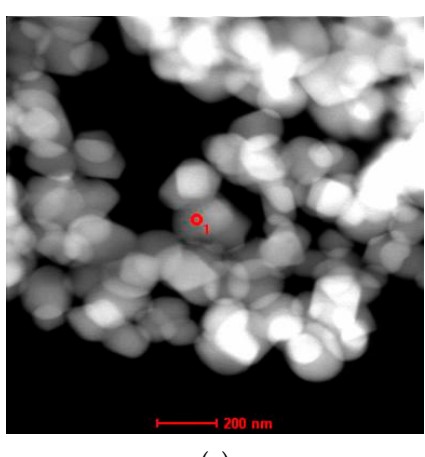 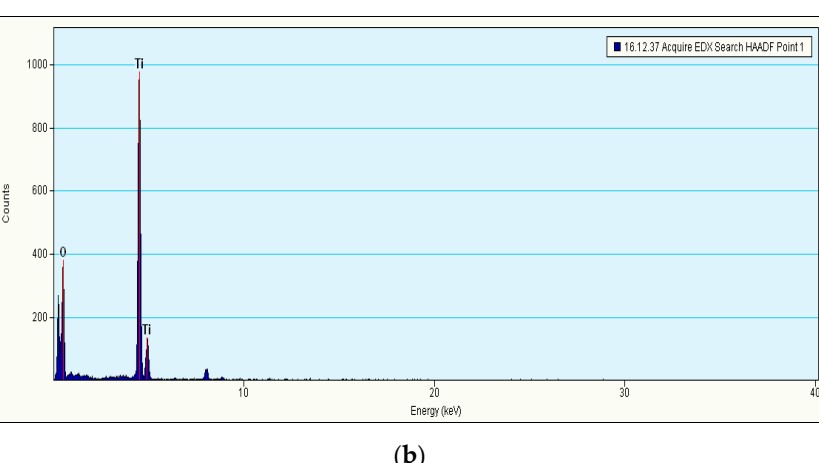

(**a**) (**b**)

**Figure 1.** (**a**) Transmission electron microscopy image of $TiO_2$; (**b**) Energy dispersive spectroscopy image of $TiO_2$.

### 4.2. Site Descriptions and Measurement Methods

On-site measurement of NOx concentrations on urban roads is difficult due to the dispersion phenomena caused by winds or other weather conditions [31]. Wind speed and direction along with temperature and relative humidity have a significant role in the atmospheric dispersion of air pollutants [32]. In order to overcome the measurement limitations on urban roads, a 400 m-long underpass section located in Daegu, Korea was

selected as a field monitoring testbed for evaluating the NOx reduction efficiency of a TiO$_2$ material applied on urban roads. The side walls in the underpass section enabled us to minimize the transportation of air pollutant in the horizontal direction.

The testbed was divided into two sections, a test section with TiO$_2$ applied and a reference section without TiO$_2$ applied. The average annual daily traffic at this testbed is approximately 13,000 vehicles. Since the test and reference section is located in a row at about 70 m distance, the effect of traffic characteristics on the NOx concentration measured in the both sections are the same. A liquid type of TiO$_2$ was applied to the surface of asphalt pavement in the test section. A customized distributor vehicle with nozzles was used in uniform coverage of TiO$_2$ on the surface of asphalt pavement. The vehicle speed was calculated to spray the target amount of TiO$_2$. With 5 km/h of vehicle speed and 3 bar of injection pressure, 0.15 kg/m$^2$ of TiO$_2$ solution was applied uniformly on asphalt pavement. The data measured in the reference section reflects on the general characteristics of NOx concentration in this testbed and will be used to estimate the NOx reduction efficiency in the test section.

Data on NOx concentration levels were measured at the test and reference section simultaneously to compare the trend of NOx concentration in both sections and to reflect the characteristics of time series data for estimating probability distributions. In order to consider the effect of vehicle emission directly on NOx concentration levels, the test probe was located 400 mm from the pavement surface. NOx concentrations, traffic volume, and meteorological information were monitored from 16 to 20 December 2019 and from 3 to 7 February 2020. The NOx concentration was measured using the chemiluminescence method, which is the set measurement standard based on the 'Framework Act on Environmental Policy' in Korea. In the chemiluminescence method, when nitrogen dioxide is produced by the reaction between nitrogen monoxide and ozone in the sample during the measurement of NOx concentration, light is generated by transitional chemiluminescence to the activated molecular ground state. This method is a way to use the degree of light generated by those chemical lightings to be calculated in proportion to the concentration of nitric oxide.

Based on the meteorological information monitored at this testbed, the average temperature ranged from −2.1 to 9.1 °C and the average amount of cloud cover ranged from 0.8 to 6.0 Okta (in meteorology, an Okta is a unit of measurement used to describe the amount of cloud cover at any given location.). Also, the average wind speed ranged from 0.9 to 4.5 m/s and wind directions were mostly northwesterly during the measurement days. Tables 1 and 2 present the characteristics of on-site data collection and weather information during the measurement days, respectively.

*4.3. Anderson-Darling Test*

NOx concentrations are generally calculated by applying a daily or time means to a reference concentration. However, the purpose of this study is to analyze the effect of TiO$_2$ on reducing NOx concentration, and there is a limit to analyzing NOx concentration changes since the NOx analysis section will not reflect changes in NOx concentration during a specific time or day. Therefore, in order to analyze the effect of TiO$_2$ on NOx concentration reduction in this study, NOx concentration was measured at a test section with TiO$_2$ applied and a reference section without TiO$_2$ applied. The data can then be analyzed by comparing how much of the NOx analysis section is included in the range of 'high', defined as no less than 0.10 ppm, 'moderate', defined as no more than 0.10 ppm, and 'low', defined as no more than 0.06 ppm. In order to utilize this analysis method, first of all, the NOx concentration in the test and reference section will be measured based on the unit time of 1 min over a certain time period. Then, probability distributions of each measured data set will be estimated using statistical inference methods. With each estimated probability distribution, the probability of NOx occurring within the NOx analysis ranges (high, moderate, low), which were defined in this study, can be analyzed by calculating the probability density function of each probability distribution.

**Table 1.** Characteristics of Data Collection.

| Items | Contents |
|---|---|
| Location | Seopyung Underpass Section in Daegu, Korea |
| Characteristics of Testbed | Road Classification: Underpass<br>Number of Lanes: 4 Lanes<br>AADT: Approximately 13,000<br>Length of Testbed: Approximately 400 m |
| Description of Testbed | A General View of Testbed<br>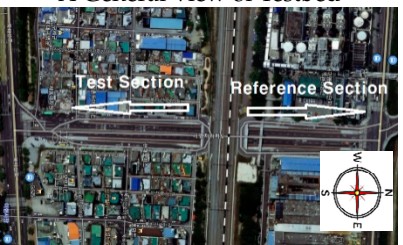<br>Test Section with $TiO_2$ applied<br>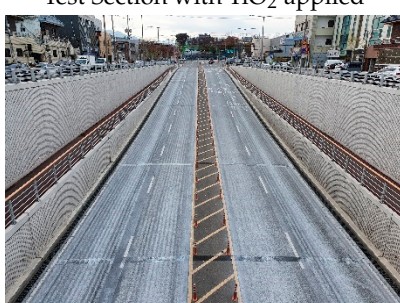<br>Reference Section without $TiO_2$ applied<br>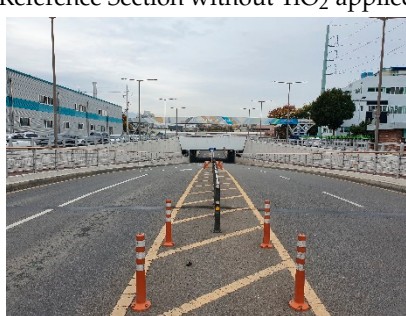 |
| Data Collection Devices | NOx Measurement Device<br>(Serinus 40 Oxides of Nitrogen Analyser)<br>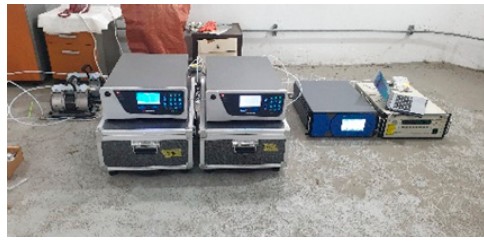 |
| Date of Data Collection | 16 to 20 December 2019<br>3 to 7 February 2020 |

**Table 2.** Weather Information of Measurement Days.

| Date | Average Temperature (°C) | Average Cloud Cover (Okta) | Average Wind Speed (m/sec) | Wind Direction |
|---|---|---|---|---|
| 16 December 2019 | 6.3 | 3.9 | 0.9 | SE |
| 17 December 2019 | 9.1 | 5.9 | 1.7 | W |
| 18 December 2019 | 7.4 | 5.3 | 3.1 | NW |
| 19 December 2019 | 4.1 | 4.3 | 4.1 | NW |
| 20 December 2019 | 2.4 | 6.0 | 2.0 | NW |
| 3 February 2020 | 3.6 | 2.0 | 2.9 | NW |
| 4 February 2020 | 1.9 | 0.8 | 1.8 | W |
| 5 February 2020 | −1.5 | 1.1 | 4.5 | NW |
| 6 February 2020 | −2.1 | 3.0 | 1.8 | NW |
| 7 February 2020 | 1.1 | 5.1 | 2.2 | NW |

First of all, to estimate the probability distribution of NOx in the test and reference sections within the NOx analysis ranges prescribed in this study, a goodness-of-fit test method can be used to determine how close the assumed probability distribution is to the empirical frequency distribution based on data obtained from the NOx concentration measured from the test and reference sections [33,34]. Therefore, this study can use the Anderson-Darling test, which tests whether a set of data follows a specific distribution, as a goodness-of-fit verification method to determine the probability distribution suitable for the NOx concentration data collected from the test and reference sections. The Anderson-Darling test is a statistical goodness-of-fit method presented by Anderson and Darling (1952) as a criterion for minimizing errors in the estimation process. This goodness-of-fit verification method is a method for verifying how much the data collected correspond to a characteristic distribution. The basic formula for the Anderson-Darling test is as follows [32]:

$$Q_n = n \int_{-\infty}^{\infty} [F_n(x) - F(x)]^2 \omega(x) \, dF(x) \tag{10}$$

where $F_n(x)$ is the empirical cumulative distribution function, $F(x)$ is the distribution function selected for analysis; and $\omega(x)$ is the weighting function.

In the basic formula, when $\omega(x) = [F(x)(F(x))]^{-1}$, this is called the AD statistic and is commonly denoted as given by [31,32]:

$$A_n^2 = n \int_{-\infty}^{\infty} \frac{\left[ [F_n(x) - F(x)]^2 \right]}{F(x)(1 - F(x))} \, dF(x) \tag{11}$$

Also, some papers, including this study, give the following as the definition of the Anderson-Darling test statistic:

$$A_n^2 \approx - \sum (2i - 1) \left[ \log\left( F\left( x_{(i)} \right) \right) + \log\left( 1 - F\left( x_{(n+1-i)} \right) \right) \right]^{-n} \tag{12}$$

In the Anderson-Darling test, the higher the goodness-of-fit between the data and specific distributions, the smaller the AD statistics calculated are. In addition, $p$-value is used to determine which of the different distributions the data fits. If the $p$-value is less than the chosen significance level of 0.05, it is common to evaluate the goodness-of-fit between the data and a particular distribution as insignificant [32]. If the amount of data collected is large, the central limit theorem can be utilized rather than the data approximate the normal distribution. Therefore, if the amount of data measured is sufficient, normality verification is not required. However, if the data sample is smaller than 30, a normality

test should be implemented to find out the distribution. If it is not a normal distribution, a nonparametric method should be applied instead of a parametric method.

## 5. Analysis of the Effect of Titanium Dioxide on Reducing Nitrogen Oxide

### 5.1. Modeling of Nitrogen Oxide Reduction for Test and Reference Sections

The on-site data collected at the testbed was utilized to estimate the effect of $TiO_2$ on reducing NOx concentration by stochastically analyzing and comparing the NOx concentration in the test and reference section after eliminating abnormalities (outliers) during data collection. Next, the probability distribution of the NOx concentration data measured at the test section and the reference section was estimated. The data collected can be analyzed based on a method of estimating the probability distribution using the probable goodness-of-fit between the measured data and a specific probability distribution. Thus, in this study, the probability of NOx concentration measured at each section was estimated using the Anderson-Darling (AD) test among the different various goodness-of-fit verification methods. In the AD test, commonly used probability distributions include normal distribution, exponential distribution, Weibull distribution, and log-normal distribution. The probability distributions appropriate for the data collected from the test and reference section can be estimated based on AD statistics and *p*-value.

According to the results of the goodness-of-fit test for the probability distribution with the Anderson-Darling test, the AD statistics for the NO concentration distributions in the test section on 16 and 17 December 2019 were 1.631 (*p*-value: <0.010) and 7.296 (*p*-value: <0.010), respectively, with both following the Weibull distribution. The AD statistics for the NOx concentration distributions in the reference section on the same dates were 2.341 (*p*-value: <0.010) and 9.501 (*p*-value: <0.010), respectively, with both following the Weibull distribution.

In addition, AD statistics for the NOx concentration distributions in the test section on 18 December 2019 and 7 February 2020 were 1.315 (*p*-value: <0.005) and 2.868 (*p*-value: <0.005), which follow the log-normal distribution. Also, the AD statistics for the NOx concentration distributions in the reference section on same dates as the test section were 0.551 (*p*-value: <0.005) and 0.869 (*p*-value: <0.010), which follow the log-normal distribution. For more detailed information on the results of the AD test, refer to Table 3 below.

**Table 3.** Results of Anderson-Darling Test in the Test and Reference Section.

| Date | | 16 December 2019 | | 17 December 2019 | | 18 December 2019 | | 19 December 2019 | | 20 December 2019 | |
|---|---|---|---|---|---|---|---|---|---|---|---|
| **Types of Section** | | **Test** | **Reference** | **Test** | **Reference** | **Test** | **Reference** | **Test** | **Reference** | **Test** | **Reference** |
| AD Statistics | Normal | 2.878 | 3.180 | 10.862 | 12.259 | 25.545 | 21.550 | 31.903 | 103.551 | 36.532 | 26.238 |
| | Expon | 61.302 | 67.425 | 31.854 | 46.012 | 47.735 | 75.952 | 50.181 | 55.544 | 13.269 | 30.366 |
| | Weibull | **1.631** | **2.341** | **7.296** | **9.501** | 9.938 | 12.408 | **14.719** | **36.982** | **10.941** | **9.586** |
| | Log-Nor | 5.339 | 4.451 | 10.367 | 11.979 | **1.315** | **0.551** | 3.660 | 2.305 | 4.448 | 3.559 |
| *p*-value | | <0.010 | <0.010 | <0.010 | <0.010 | <0.005 | <0.005 | <0.005 | <0.005 | <0.005 | <0.005 |
| Date | | 3 February 2020 | | 4 February 2020 | | 5 February 2020 | | 6 February 2020 | | 7 February 2020 | |
| **Types of Section** | | **Test** | **Reference** | **Test** | **Reference** | **Test** | **Reference** | **Test** | **Reference** | **Test** | **Reference** |
| AD Statistics | Normal | 22.981 | 20.309 | 47.827 | 64.752 | 34.226 | 26.714 | 18.662 | 92.621 | 22.921 | 15.767 |
| | Expon | 28.169 | 49.699 | 18.046 | 24.871 | 47.036 | 61.648 | 67.007 | 85.200 | 29.684 | 55.967 |
| | Weibull | 7.921 | 10.634 | 123.97 | 17.898 | 11.544 | 11.191 | 7.906 | 41.041 | 7.965 | 6.796 |
| | Log-Nor | **0.951** | **1.264** | **3.992** | **2.458** | **0.242** | **0.630** | **0.595** | **1.328** | **2.868** | **0.869** |
| *p*-value | | <0.010 | <0.005 | <0.005 | <0.005 | <0.010 | <0.010 | <0.010 | <0.005 | <0.005 | <0.010 |

Bold characters are the AD statistics of accepted distributions; Types of Section: Test→A segment of road with $TiO_2$ applied, Reference→A segment of road without $TiO_2$ applied.

The probability density function of each estimated probability distribution was computed to analyze the reduction effect of $TiO_2$ on NOx concentration. This was calculated using the Anderson-Darling test with the probability distributions of each day measured and the two types of sections. The probability density function can be analyzed by comparing how much of the NOx concentration is included in the range of above 0.10 ppm for

'high', between 0.06 and 0.10 ppm for 'moderate', and below 0.06 ppm for 'low'. Figure 2 shows computational results of probability density function of each estimated distribution.

### 5.2. Evaluation of TiO₂ Effect on NOx Reduction

Figure 3 shows a comparison of the daily mean NOx concentration data measured in the test and reference sections. This figure reveals that NOx concentration in the test section is generally lower than that in the reference section, indicating that use of TiO₂ is capable of reducing NOx concentrations on urban roads. Table 4 presents the summary of analysis results for the NOx concentration reduction effect with the descriptive statistics of measured data. On 16 December 2019, means of the NOx concentration in the test and reference section were 0.213 ppm and 0.226 ppm, and the NOx concentrations in each section were recorded to be between 0.043~0.504 ppm and 0.037~1.059 ppm, respectively. Also, from 17 to 20 December 2019 and 3 to 7 February 2020, means of NOx concentration in the test section ranged from 0.047 to 0.218 ppm and in the reference section, it ranged from 0.077 to 0.219 ppm. During the same time period mentioned above, the NOx concentrations are in the range between 0.002 to 0.789 ppm in the test section and 0.031 to 3.599 ppm in the reference section.

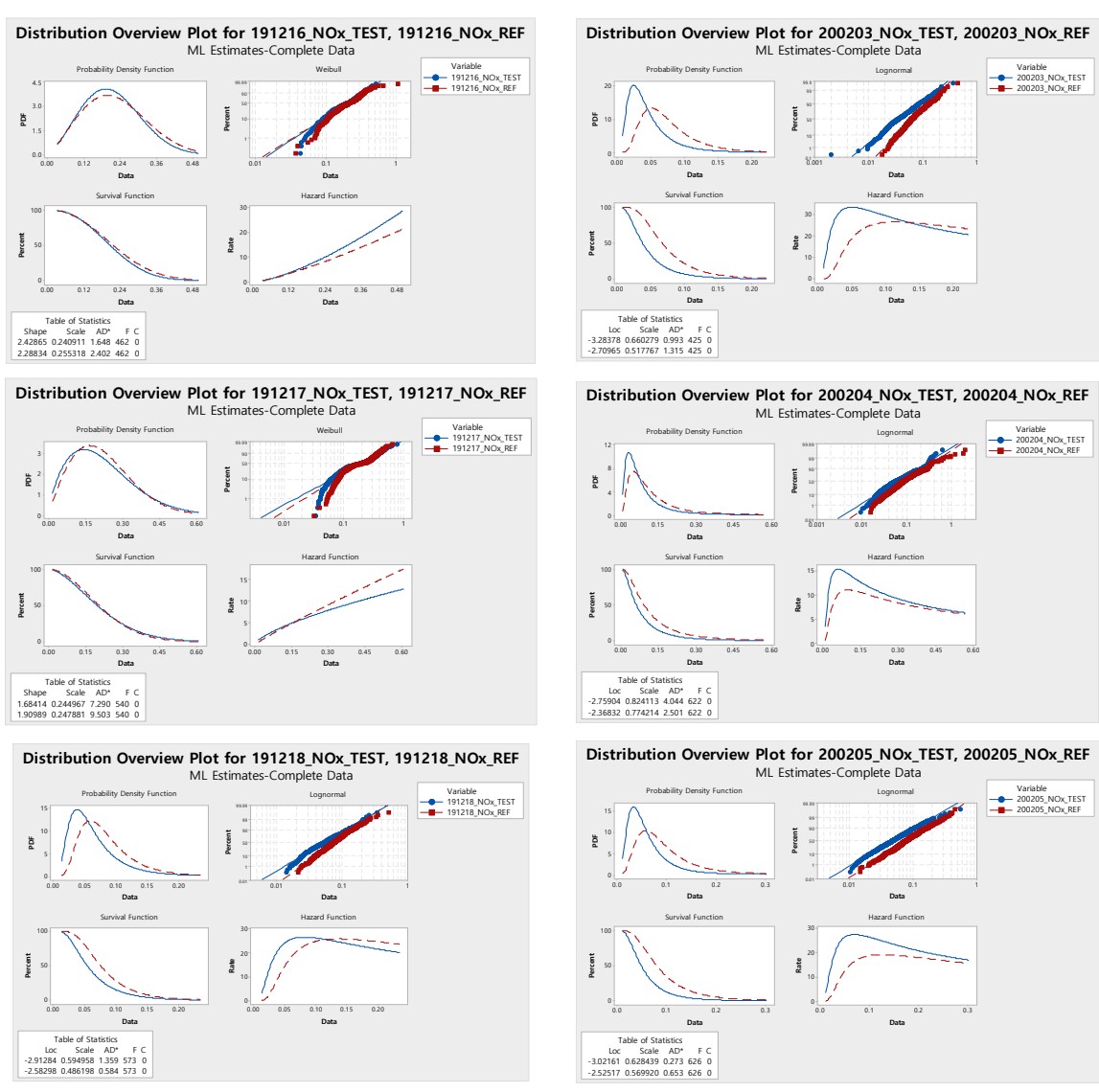

**Figure 2.** *Cont.*

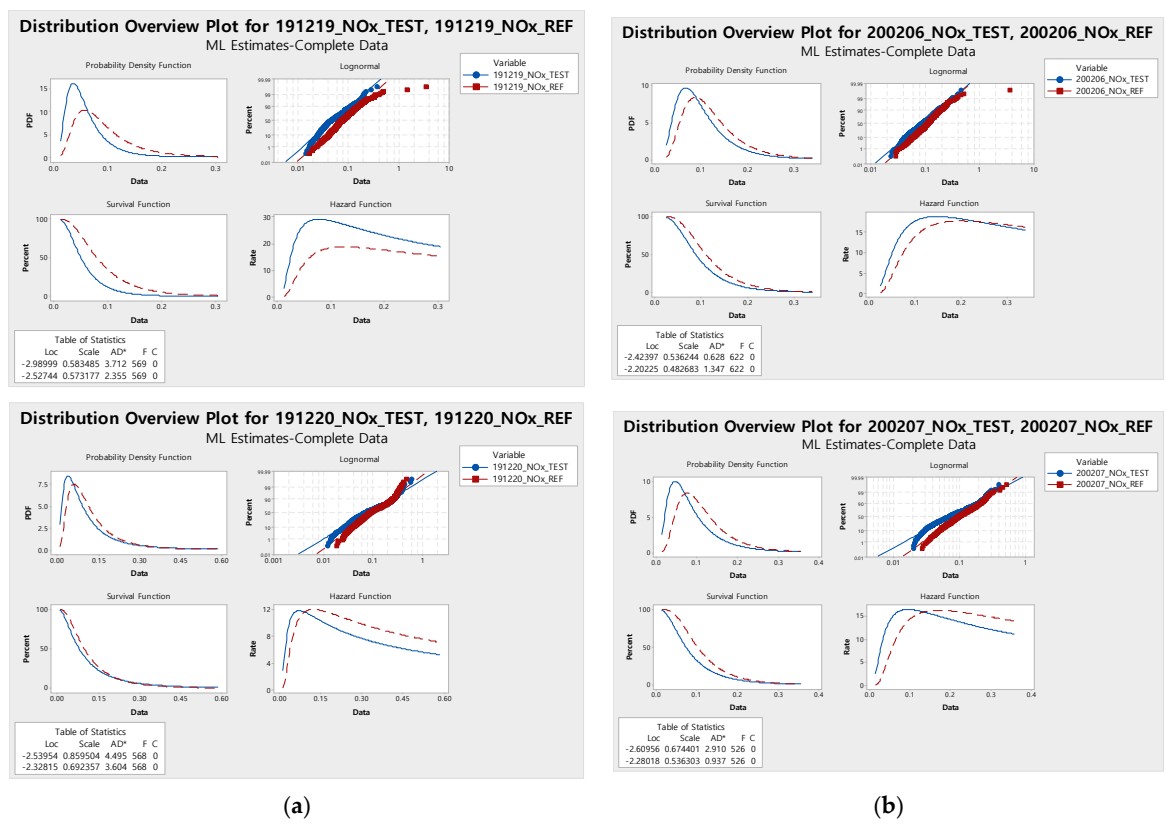

**Figure 2.** Probability distributions and probability density functions of the Anderson-Darling test for the test and reference sections: (**a**) Date of measurement: 6 to 20 December 2019; (**b**) Date of measurement: 3 to 7 February 2020.

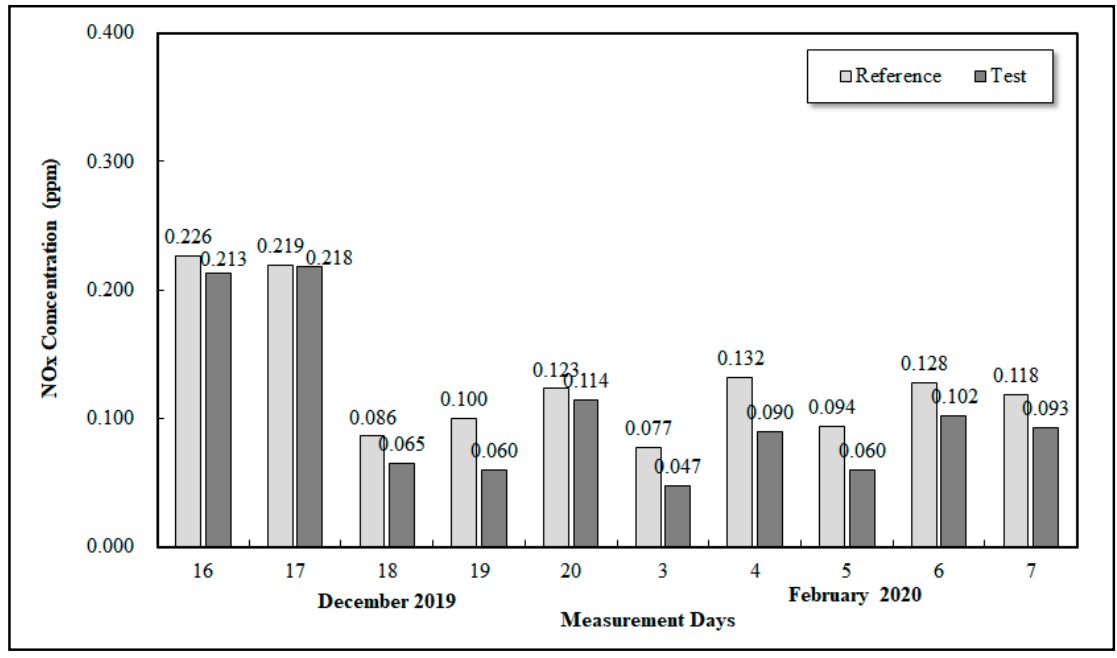

**Figure 3.** Means of NOx concentration on the test and reference section in measurement days.

**Table 4.** Analysis Results of Characteristics of NOx in the Test and Reference Section.

| Date | Type of Sections | Number of Data | Mean (ppm) | Standard Deviation (ppm) | Max (ppm) | Min (ppm) | Distribution | Area from Probability Density Function (Probability of Inclusion in Standard Section) | | |
|---|---|---|---|---|---|---|---|---|---|---|
| | | | | | | | | High (>0.01 ppm) | Moderate (0.06~0.10 ppm) | Low (<0.06 ppm) |
| 16 December 2019 | Test | 426 | 0.213 | 0.095 | 0.504 | 0.043 | Weibull | 88.9% | 7.8% | 3.3% |
| | Reference | 426 | 0.226 | 0.103 | 1.059 | 0.037 | Weibull | 89.0% | 7.4% | 3.6% |
| 17 December 2019 | Test | 540 | 0.218 | 0.135 | 0.789 | 0.033 | Log-normal | 80.2% | 10.9% | 8.9% |
| | Reference | 540 | 0.219 | 0.122 | 0.656 | 0.031 | Log-normal | 83.8% | 9.7% | 6.5% |
| 18 December 2019 | Test | 573 | 0.065 | 0.044 | 0.337 | 0.014 | Log-normal | 15.3% | 28.1% | 56.6% |
| | Reference | 573 | 0.086 | 0.049 | 0.518 | 0.021 | Log-normal | 28.2% | 40.0% | 31.8% |
| 19 December 2019 | Test | 569 | 0.060 | 0.042 | 0.378 | 0.015 | Log-normal | 11.9% | 26.2% | 61.9% |
| | Reference | 569 | 0.100 | 0.164 | 3.466 | 0.017 | Log-normal | 34.7% | 34.4% | 30.9% |
| 20 December 2019 | Test | 568 | 0.114 | 0.102 | 0.592 | 0.012 | Log-normal | 39.1% | 23.4% | 37.5% |
| | Reference | 568 | 0.123 | 0.087 | 0.484 | 0.019 | Log-normal | 48.5% | 27.3% | 24.2% |
| 3 February 2020 | Test | 425 | 0.047 | 0.037 | 0.361 | 0.002 | Log-normal | 6.9% | 16.9% | 76.2% |
| | Reference | 425 | 0.077 | 0.047 | 0.437 | 0.018 | Log-normal | 21.6% | 36.4% | 42.0% |
| 4 February 2020 | Test | 622 | 0.090 | 0.085 | 0.611 | 0.009 | Log-normal | 29.0% | 23.6% | 47.4% |
| | Reference | 622 | 0.132 | 0.160 | 2.010 | 0.016 | Log-normal | 46.6% | 25.1% | 28.3% |
| 5 February 2020 | Test | 626 | 0.060 | 0.047 | 0.550 | 0.010 | Log-normal | 12.6% | 24.4% | 63.0% |
| | Reference | 626 | 0.094 | 0.061 | 0.452 | 0.015 | Log-normal | 34.8% | 34.5% | 30.7% |
| 6 February 2020 | Test | 622 | 0.102 | 0.060 | 0.456 | 0.023 | Log-normal | 41.0% | 35.6% | 23.4% |
| | Reference | 622 | 0.128 | 0.152 | 3.599 | 0.028 | Log-normal | 58.2% | 31.5% | 10.3% |
| 7 February 2020 | Test | 526 | 0.093 | 0.066 | 0.400 | 0.020 | Log-normal | 32.4% | 29.4% | 38.2% |
| | Reference | 526 | 0.118 | 0.068 | 0.522 | 0.028 | Log-normal | 51.7% | 32.3% | 16.0% |

Types of Section: Test→A segment of the road with $TiO_2$ applied, Reference→A segment of the road without $TiO_2$ applied.

The NOx concentrations in the testbed appeared in a rather wide range due to various factors such as atmospheric environment, weather conditions, and traffic characteristics during the measurement data. Moreover, the overall high level of NOx concentration in this study can be attributed to the measurement of NOx concentration at a close distance from passing vehicles.

Based on the analysis of the NOx concentration reduction effect of $TiO_2$ using probability density functions, the occurrence probability of NOx concentration in the test section on 18 December 2019 was 'high' at 15.3%, 'moderate' at 28.1%, and 'low' at 56.6%. The occurrence probability of NOx concentration in the reference section on the same day was 'high' at 28.2%, 'moderate' at 40.0%, and 'low' at 31.8%. On 19 December 2019, the occurrence probability of NOx concentration in the test section was 'high' at 11.9%, 'moderate' at 26.2%, and 'low' at 61.9%. In the reference section for the same date, the occurrence probability was 'high' at 34.7%, 'moderate' at 34.4%, and 'low' at 30.9%. Also, the occurrence probability of NOx concentration in the test section on December 20th was 'high' at 39.1%, 'moderate' at 23.4%, and 'low' at 37.5%, while the occurrence probability of NOx concentration in the reference section was 'high' at 48.5%, 'moderate' at 27.3%, and 'low' at 24.2%. In addition, the occurrence probability of NOx concentration from 3 to 7 February 2020 follow a similar trend to the results from 18 to 20 December 2019.

However, the results of analysis on 16 and 17 December 2019 showed that the occurrence probability of NOx concentration in the test section was 88.9% and 80.2% in 'high', 7.8% and 10.9% in 'moderate', 3.3% and 8.9% in 'low', respectively. The occurrence probability of NOx concentration in the reference section on the two dates was analyzed as 89.0% and 83.8% in 'high', 7.4% and 9.7% in 'moderate', 3.6% and 6.5% in 'low'. This shows a slightly different result compared to other dates of measurement and could have been due to the effects of very cloudy weather conditions on the measurement of NOx.

Comparing the occurrence probability of NOx concentration between the test and reference sections, the average occurrence probability of NOx concentration in the reference section in the 'high' range was about 28.1% higher than that of the test section, and in the case of 'moderate,' the average occurrence probability of the NOx concentration in the reference section was about 18.8% higher than that of the test section. On the other hand, in the case of the 'low' range, the average probability of NOx concentration in the test section was about 46.2% higher than that of the reference section. This fact indicated that the NOx concentration in the test section was statistically lower comparing to the reference section.

It is well known that it is difficult to validate the reduction effect of $TiO_2$ using field experiments due to a large number of influencing variables such as wind speed, vehicle type, humidity, and temperature. However, it can be concluded from on-site measurement and statistical analysis that applying $TiO_2$ on asphalt pavement surfaces is capable of reducing the NOx concentration through the photocatalytic reaction with sufficient solar radiation.

## 6. Summary and Conclusions

In this study, in order to analyze the effect of $TiO_2$ on reducing NOx concentration, a full-scale testbed was constructed at an underpass section. A liquid type of $TiO_2$ was sprayed onto the surface of asphalt pavement in the test section, whereas no $TiO_2$ was applied on the reference section for comparison purposes. NOx concentration was measured and collected at a one-minute interval in the two sections at the same time for 10 days. The reduction effects on NOx concentration were analyzed by calculating the occurrence probability of the NOx concentration in the test and reference sections using the Anderson-Darling test, which is a statistical verification method for the collected data.

According to the analysis results, most of the NOx concentration data measured in the test section were probabilistically lower than those in the reference section. As a result, the average occurrence probability of the NOx concentration in the reference section at 'high' and 'moderate' was about 28.1% and 18.8% higher than that of the test section, respectively. In the 'low' range, the average probability of NOx concentration in the test

section was about 46.2% higher than that of the reference section. These results showed that the photochemical reaction of $TiO_2$ material applied on asphalt pavement was effective due to good weather conditions on the days of measurement.

In this study, an analysis of the reduction effect on NOx concentration from applying $TiO_2$ onto road facilities is expected to be used in future studies for removing pollutants and gas precursors on urban roads. The on-site application of $TiO_2$ material at various types of road sections is required to validate the NOx reduction efficiency by considering various influencing parameters. In the future, it is necessary to conduct long-term monitoring of NOx concentration changes to evaluate the degradation of durability and reduction performance of the $TiO_2$ material utilized on the testbed.

**Author Contributions:** Conceptualization, S.-H.L. and H.-M.P.; methodology, S.-H.L.; software, S.-H.L.; validation, H.-M.P. and S.-H.L.; formal analysis, S.-H.L.; investigation, J.-W.L. and M.-K.K.; resources, H.-M.P., J.-W.L. and M.-K.K.; writing—original draft preparation, S.-H.L. and J.-W.L.; writing—review and editing, H.-M.P.; visualization, S.-H.L.; supervision, H.-M.P.; project administration, H.-M.P.; funding acquisition, H.-M.P. All authors have read and agreed to the published version of the manuscript.

**Funding:** This research received no external funding.

**Institutional Review Board Statement:** Not applicable.

**Informed Consent Statement:** Not applicable.

**Data Availability Statement:** Data sharing not applicable.

**Acknowledgments:** The authors are grateful for the financial support for the research from the Korea Institute of Civil Engineering and Building Technology (KICT).

**Conflicts of Interest:** The authors declare that there is no conflict of interests regarding the publication of this article.

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
