# Peer review of "An Analysis on the Effectiveness of Nitrogen Oxide Reduction from Applying Titanium Dioxide on Urban Roads Using a Statistical Method"

_atmosphere, doi:10.3390/atmos12080972_

Round 1

Reviewer 1 Report

The paper analyses the data obtained in a real experiment and compares them with data from a statistical method.

My decision is "Reconsider after major revision (control missing in some experiments)" due to a more information is needed to understand some experiment.

First of all, when work with a statistical method is important the number of data that you collect, in this paper the authors didn´t comment this information, please, could you comment on it?.

Second, on the selection of place where the authors collect the data, the authors aren´t clear in the text description, fortunately, they include a picture on table 1 to clarify the situation.

On the text, the authors have repeatedly commented that the decontamination capacity of the system depends on many factors, including environmental, traffic, etc., one of those parameters is the wind direction, this parameter, is important for how the authors have define the test and reference sections. If the authors know the wind direction during the experiment days, please comment it. If the authors don´t know this information, please search what is the most common wind direction in this area, maybe the local authorities responsible of the tunnel maintenance have this information.

In other things, on page 4 from line 165 to line 189 there are several mistakes relative to the negative charge of compounds (must be superscript) and other must be radical (include the point that indicates it).

On page 9 from line 319 to line 326 the information that comment is possible to read at the table and it isn´t necessary to comment it in the text. Same thing on page 11 from line 338 to line 354.

Author Response

I attached the answer file. Please see the file. Thank you.

Reviewer 2 Report

This manuscript presents an analysis on the effectiveness of nitrogen oxide reduction from applying titanium dioxide on urban roads using the statistical method. In this paper, the application of TiO2 as air purification catalysts on urban roads has been analyzed. The authors should clearly explain the choice of TiO2 as a photocatalyst for their experiments. References should be updated with more recent reports about novel methods and photocatalysts for air purification of road. Please explain what is novel and what are the differences in this work in the introduction. The authors should clearly justify the choice of their methods and the advantages of their work.

The SEM or TEM images of the TiO2 particles should be provided for the information of shape and size distribution of the used photocatalyst. The spray method and the loading mass of TiO2 on the urban road should be presented in detail. English in some parts needs to be polished.

After major revisions and clear explanations, this manuscript can be published in the Atmosphere.

Author Response

(The authors gave the same response as above.)

Reviewer 3 Report

Researchers are investigating an interesting area in road transport that add emission reduction to pavement design and material development. Experiment was conducted to collect primary data that made a contribution to the literature. However, several weakness need to be addressed in the manuscript:

  1. It is quite confusing to introduce other airborne emissions, e.g. CO, PM, to this study while NOx is the only measurement.
  2. Details of the experimental road are needed – is there a difference in vehicle mix, speed, etc. between the test and reference section? Photographs show that the two sections are in different street cannon with different land use by roadside. Authors need to ensure fair comparison is made.
  3. From impact point of view, details of the asphalt pavement surface need to be revealed in the manuscript – are the test and reference section have same type of surface, year of construction?
  4. Results indicated the NOx reduction, compared to the reference section, is more significant after 7 weeks. Authors need to clarify and if this is the case, will contradict with what the literature says then further discussion is worthwhile.
  5. Info is repeated in texts, tables and figures.

Author Response

(The authors gave the same response as above.)

Round 2

Reviewer 1 Report

The answers to the authors are in the attach document. The authors clarify all my question except one, which is relative to the wind direction. Tha author clarify teh wind direction but on the picture the are not a compass to know how affect this direction to the test. 

Author Response

Thank you for valuable comments and suggestion for this paper.

Also, I added the compass on the picture in Table 1 according to your opinion. 

Reviewer 2 Report

The authors adjust well according to most of my comments and show the new TEM image. This manuscript can be published now in the Atmosphere.

Author Response

Thank you for your review.